

# Brief communication: Drought likelihood for East Africa

**Hui Yang**[1,2] **and Chris Huntingford**[2]

[1]Department of Ecology, School of Urban and Environmental Sciences, Peking University, Beijing, 100871, PR China
[2]Centre for Ecology and Hydrology, Wallingford, Oxfordshire, OX10 8BB, UK

**Correspondence:** Hui Yang (yang_hui@pku.edu.cn)

**Abstract.** TS1 TS2 The East Africa drought in autumn of year 2016 caused malnutrition, illness and death. Close to 16 million people across Somalia, Ethiopia and Kenya needed food, water and medical assistance. Many factors influence drought stress and response. However, inevitably the following question is asked: are elevated greenhouse gas concentrations altering extreme rainfall deficit frequency? We investigate this with general circulation models (GCMs). After GCM bias correction to match the climatological mean of the CHIRPS data-based rainfall product, climate models project small decreases in probability of drought with the same (or worse) severity as 2016 ASO (August to October) East African event. This is by the end of the 21st century compared to the probabilities for present day. However, when further adjusting the climatological variability of GCMs to also match CHIRPS data, by additionally bias-correcting for variance, then the probability of drought occurrence will increase slightly over the same period.

## 1 Introduction

TS3 Historical rainfall estimated by Climate Hazards Group InfraRed Precipitation with Station data (CHIRPS; Funk et al., 2015) shows that, during August to October (ASO) of 2016, large parts of Somalia, Ethiopia and Kenya (black rectangle, Fig. 1a) had a reduction of 40 % or more in rainfall compared to a baseline ASO period 1981–2015. For this region, the spatial average of monthly rainfall during ASO of 2016 lies at least 1 CE1 standard deviation (SD) below the climatological mean of the other years (Fig. 1b). The year of 2016 is the driest year in the past four decades. Other years with rainfall at least 1 SD below the climatological mean during 1981–2015 are 1983–1986, 1990 and 1991. We

concentrate on East Africa, as this region experienced particularly poor harvest and famine was widely reported during 2016 (noting that many regions outside the black rectangle of Fig. 1a also experienced major rainfall deficits in 2016). East Africa is especially vulnerable to the impacts of drought (DEC, 2017) TS4. The region has long experienced widespread poverty and high levels of food insecurity (Von Grebmer et al., 2016). The high dependence of its population on rain-fed agriculture, sometimes in tandem with political changes, exacerbates the impacts of droughts (Love, 2009; Masih et al., 2014).

To assess any influence of increasing atmospheric greenhouse gas concentrations, we use monthly rainfall data from 37 general circulation model (GCM) simulations for the historical period and for the high-emission future scenario RCP8.5. These are from the Coupled Model Intercomparison Project Phase 5 (CMIP5, Taylor et al., 2012). A summary of the main characteristics of the models is provided in Table S1 in the Supplement. A bias correction with two postprocessing steps is applied to the GCM precipitation estimates. We first calculate modelled and CHIRPS-based mean ASO rainfall estimates over the East Africa region (set as within black rectangle, Fig. 1a) and during the period 1981–2015. The GCM precipitation mean ASO estimates, both past and future, are corrected for each model year by a GCM-specific mean correction factor. This factor is a ratio of the climatological mean of each GCM to that of the CHIRPS product as

$$x^{\mu}_{\text{corr},i,j} = x_{\text{model},i,j} \times \frac{\mu_{\text{obs}}}{\mu_{\text{model},j}}. \qquad (1)$$

Here $x_{\text{model},i,j}$ and $x^{\mu}_{\text{corr},i,j}$ are, respectively, model simulated and mean bias-corrected ASO precipitation data of the $i$th year ($i = 1, 2, \ldots, 31$) for the $j$th GCM ($j = 1, 2, \ldots, 37$).

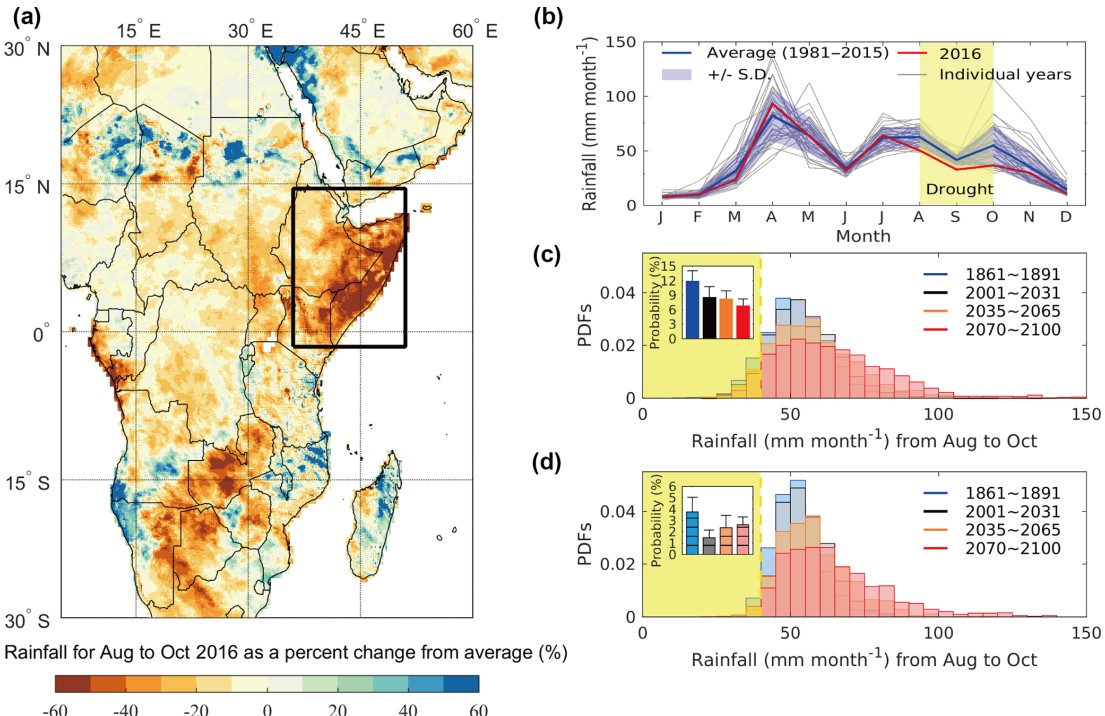

**Figure 1. (a)** Black rectangle is the location of the study region (14.5° N–1.5° S, 36–51° E). Plotted is mean rainfall for 2016 and months August to the end of October (ASO), presented as relative change (as %) to the long-term average ASO values (1981–2015). Values based on CHIRPS precipitation product. **(b)** CHIRPS-based monthly total rainfall (mm month$^{-1}$) over study region (panel **a**; land within black rectangle) for years 1981 to 2016. Year 2016 is red, other years are individual grey lines, and multi-year average (not including 2016) is the blue line. Blue shading is ± 1 SD of monthly rainfall across years 1981–2015. The drought event of 2016 is defined as the three consecutive months of ASO (yellow shading), and noting rainfall in that year is below blue shading in those years. **(c)** CMIP5-based probability density functions (PDFs; binned to 5 mm month$^{-1}$ intervals) of mean ASO rainfall for periods 1861–1891 (blue), 2001–2031 (black), 2035–2065 (orange) and 2070–2100 (red). Each curve corresponds to combined estimates from 37 CMIP5 GCMs, with each GCM forced by historical emissions and RCP8.5 future scenario. Individual GCM mean bias correction is based on the CHIRPS precipitation product. Yellow shading is mean ASO rainfall less than 40 mm month$^{-1}$, which is the CHIRPS 2016-based threshold (mean of ASO, red curve in panel **b**). Inset shows probabilities of mean rainfall of ASO falling below the threshold for the same modelled periods (colours match those of curves and legend). The error bars are 2 SD (estimated via bootstrapping 80 % replications from the 37 GCM precipitation data for the 31-year periods). **(d)** Same as **(c)**, but based on the mean- and variance-corrected GCM rainfall estimates.

$\mu_{\mathrm{obs}}$ and $\mu_{\mathrm{model},j}$ are the observed and GCM-specific time mean (i.e. average across indices $i$) of ASO rainfall estimates during the period 1981–2015. Second, we then adjust the mean-corrected data from Eq. (1), such that they further are corrected to have an identical SD to the CHIRPS product whilst maintaining the bias correction for the mean. This gives bias-corrected estimates $x_{\mathrm{corr},i,j}^{\mu,\sigma}$ as

$$x_{\mathrm{corr},i,j}^{\mu,\sigma} = \left(x_{\mathrm{corr},i,j}^{\mu} - \overline{x_{\mathrm{corr},j}^{\mu}}\right) \times \left(\frac{\sigma_{\mathrm{obs}}}{\sigma_{\mathrm{corr},j}^{\mu}}\right) + \overline{x_{\mathrm{corr},j}^{\mu}}, \qquad (2)$$

where $\overline{x_{\mathrm{corr},j}^{\mu}}(= \mu_{\mathrm{obs}})$ is the 31-year average of mean bias-corrected data from Eq. (1). $\sigma_{\mathrm{obs}}$ and $\sigma_{\mathrm{corr},j}^{\mu}$ are SDs of the ASO rainfall estimates during the period 1981–2015 from observations and from the mean bias-corrected precipitation data created by Eq. (1). The adjustment of spread of rainfall distribution to match measurements is an important addi-

tional procedure to further constrain GCM estimates (Sippel et al., 2016; Jeon et al., 2016; Angélil et al., 2017). Together Eq. (1) ensures all GCMs have the CHIRPS-based mean, and with Eq. (2) also CHIRPS-based SD for the period 1981–2015. Histograms of bias-corrected mean ASO rainfall are presented in Fig. 1c for mean bias correction, and in Fig. 1d for mean and SD bias correction. These are derived from 37 GCMs, and for four 31-year periods (representing pre-industrial, present-day and two future periods as marked). All GCMs are considered equally plausible.

We estimate the probability, in any year, of mean rainfall being less than 40 mm per month and during August–October period. This threshold is 25 % less than the climatological ASO mean and is the ASO CHIRPS estimate of mean rainfall level in the year 2016 drought (red curve within yellow highlight, Fig. 1b). For the mean-corrected GCM estimates, we compare (inset, Fig. 1c) the modelled period 1861–1891,

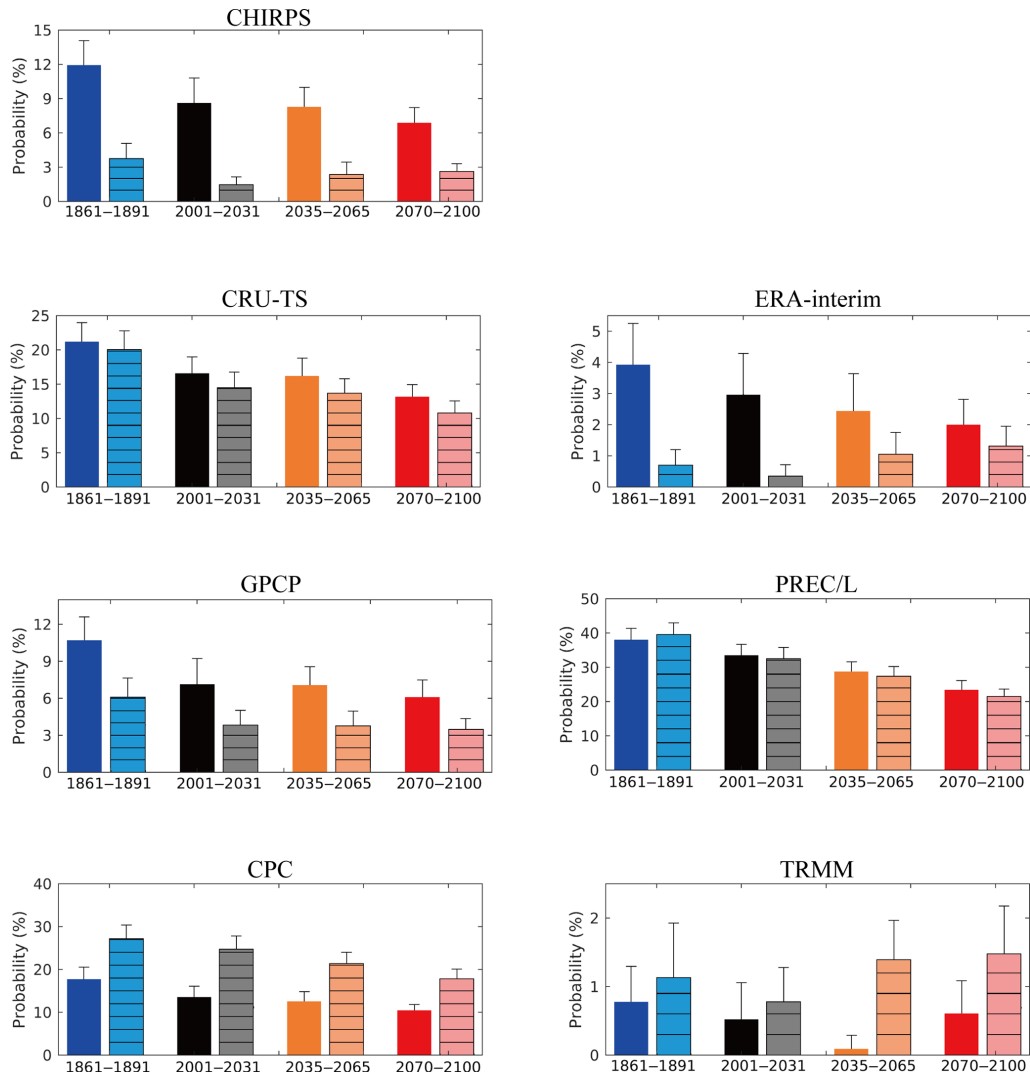

**Figure 2.** CMIP5 GCM-based histograms of probabilities of mean ASO rainfall falling below year 2016-based threshold values. This is for different time periods and for different observation-based precipitation product of CHIRPS, CRU-TS, ERA-interim, GPCP, PREC/L, CPC and TRMM. Shown for years 1861–1891 (blue), 2001–2031 (black), 2035–2065 (orange) and 2070–2100 (red), and using GCM simulations corresponding to historical and RCP8.5 estimates. Individual GCM projections are bias-corrected by the (panel-specific) precipitation product. These data are combined to give single overall probabilities across the 37 GCMs sampled. The histogram bars without horizontal hatching (left) are for the mean-corrected GCM precipitation estimates. The bars with hatching (right) are for the mean- and variance-corrected GCM estimates. The error bars are 2 SD (estimated via bootstrapping 80 % replications from the 37 GCM precipitation data for the 31-year periods). Data in the CHIRPS panels repeat those of the insets of Fig. 1c and d.

representative of pre-industrial, with present day (period 2001–2031), and find this probability decreases slightly from 11.9 % (SD ± 1.1 %) to 8.6 % (SD ± 1.1 %). The 1 SD values are estimated via bootstrapping with 80 % replications from the 37 GCM precipitation data and for the 31-year periods. These trends continue, giving probabilities 8.3 % (±0.9 %) and 6.9 % (±0.7 %) for periods 2035–2065 and 2070–2100 respectively. However, for the mean- and variance-corrected GCM estimates (Fig. 1d and inset), we find the probability of East African drought is smallest at present (1.5 % ± 0.3 %, period 2001–2031). This probability becomes larger in the

future, giving values of 2.4 % (±0.6 %) and 2.6 % (±0.4 %) for periods 2035–2065 and 2070–2100 respectively. Hence we find that additionally accounting for model biases in the variance, GCM distributions suggest a potential to significantly alter the predictions of drought events occurrence over East Africa, and for higher extreme frequency as the 21st century progresses.

Large uncertainty in the observation-based precipitation products has been well reported (Angélil et al., 2016), and so we additionally use six other precipitation estimates (CRU-TS, ERA-interim, GPCP, PREC/L, CPC and TRMM) to bias-

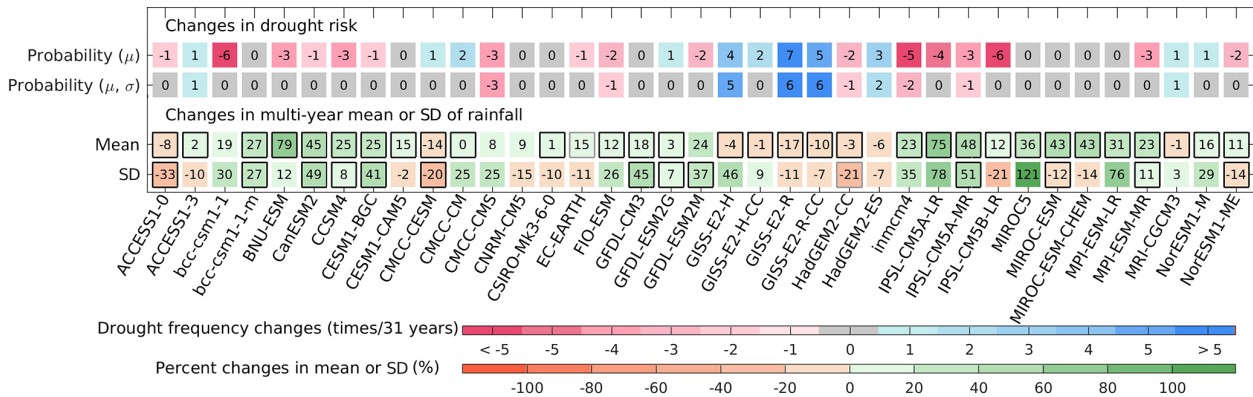

**Figure 3.** Rows 1 and 2 are changes in drought frequency (times per 31 years; top colour bar), for two methods of bias removal (mean-corrected only marked as "$\mu$" and mean and SD corrected as "$\mu, \sigma$"). This is for each of the 37 GCMs as labelled, and comparing the difference between the present period of 2001–2031 and period 2070–2100. GCM bias correction and 2016 ASO rainfall threshold are from the CHIRPS rainfall product. Rows 3 and 4 show the GCM-based changes in multi-year mean and SD of ASO rainfall respectively, and between the same periods as top rows (bottom colour bar). Black borders indicate statistically significant differences in the 31-year rainfall mean between these two periods (row 3, $t$ test, with $P < 0.05$) and significant difference in SD of GCM projections (row 4, $F$ test, with $P < 0.05$). Light grey borders in row 3 and row 4 indicate statistically significant difference at 5–10 % significance level ($0.05 \leq P < 0.1$). Values in the third and fourth rows are the percentage changes in 31-year mean and SD of rainfall as $\left[ \left( \overline{x^{\mu,\sigma}_{\text{corr},2070-2100,j}} / \overline{x^{\mu,\sigma}_{\text{corr},2001-2031,j}} \right) - 1 \right] \times$ 100 % and $\left[ \left( \sigma^{\mu,\sigma}_{\text{corr},2070-2100,j} / \sigma^{\mu,\sigma}_{\text{corr},2001-2031,j} \right) - 1 \right] \times 100\,\%$, respectively. Here overbar is time-averaging over period of interest.

correct GCM estimates. The probability of drought occurrence is based on estimates of ASO rainfall in 2016 for each individual dataset (values in Table S2). There are substantial differences between these values. We use each of these extra datasets to repeat the bias correction of every GCM by the same algorithm of Eqs. (1) and (2) but now with new data-specific $\mu$ and $\sigma$ values. These $\mu$ and $\sigma$ quantities are given in Table S2. In Fig. 2 (first panel) we reproduce the insets of Fig. 1c (no hatching) and Fig. 1d (hatching) for CHIRPS, and then for the six other precipitation products (next six panels). Consistent with the conclusions based on the CHIRPS product only, the results from the other rainfall products also show that the probability of drought occurrence in East Africa has decreased slightly from pre-industrial to present day, and irrespective of whether variance adjustment has occurred (Fig. 2, all blue and black bars, with and without hatching). Future projections of drought likelihood do, however, vary depending on precipitation product used. For the mean-corrected GCM estimates, six out of seven rainfall product-corrected GCM projections give a slight decrease in drought occurrence likelihoods by the end of the 21st century. The exception is the TRMM-corrected GCMs, which suggest the drought probability will increase slightly by 2070–2100 and relative to the present day. For the likely more appropriate mean- and variance-corrected GCM estimates, then relative to present-day levels the GCM estimates corrected to the CHIRPS, ERA-interim, and TRMM products give an increase in future drought occurrence probability. However GPCP-, PREC/L- and CPC-corrected GCM estimates suggest the probability of drought occurrence will

slightly decrease. This divergence is due to the strong differences in the climatological mean, SD and year 2016 ASO rainfall levels among the different precipitation products (Table S2).

As a sensitivity study, we also perform a bias correction based on each precipitation product but for the full ensemble of 37 GCM estimates together. That is, we combine all GCM present-day estimates into one single vector and calculate single overall $\mu$ and $\sigma$ values. All seven precipitation datasets are used to repeat the bias correction with similar methods to Eqs. (1) and (2). This approach implies that the probability of drought occurrence in East Africa has decreased slightly from pre-industrial to the end of the 21st century, regardless of whether variance has been corrected (Fig. S1). However this approach should be viewed with caution, as making single bias corrections for all the GCMs combined neglects model differences, which are known to be large in precipitation projections (Collins et al., 2013).

Our results are broadly consistent with the recent analysis of Ethiopian drought projections by Philip et al. (2017), who also use observations to reduce the model uncertainty in GCM projections. They project future changes in drought by the use of only GCMs which can reproduce well the observed distribution of February to September climatological rainfall. They find that under RCP8.5 scenario there is no significant change in the likelihood of 2015 Ethiopian drought event. Although it is in many regards logical to exclude models that do not perform well for modelling the contemporary period, our approach is possibly more cautious. This is because there always remains a concern that a rejected model may hold

important information about expected future changes, even if having strong biases in modelling the present day. Nevertheless, as a further sensitivity study, we also apply the same method as Philip et al. (2017) for our study region. This is with the CHIRPS dataset, and we place our findings in Fig. S2. The probability of 2016 ASO drought is based on rainfall projections from three models (i.e. CMCC-CM, GFDL-ESM2G and MPI-ESM-MR). They are the only models that match the climatology from CHIRPS product when using a Kolmogorov–Smirnov test, and at a significance level of 0.1. The results are generally consistent with both the mean- and variance-corrected GCM results of Fig. 1d. That is, they indicate that the probability of drought occurrence in East Africa may increase slightly from present towards the end of the 21st century.

The multi-model ensemble forecast, corrected by the CHIRPS rainfall product and merging the individual forecasts with equal weights, shows that the East African mean ASO rainfall for 2070–2100 will increase significantly, compared with the present period 2001–2031 (Fig. 1d). It is these general increases that, even in conjunction with larger future distribution spreads, imply no massive increase of drought occurrence probability (insets, Fig. 1c, d). However in Fig. 3, we present for the individual models as well. Shown are changes in numbers of years of mean ASO rainfall falling below 40 mm per month. This is for the individual GCMs bias-corrected against present-day mean ASO rainfall only (top row), or additionally against SD (second row). Both rows illustrate some individual GCMs project quite substantial changes. We also show individual model percentage changes in mean (third row) and SD (bottom row) of ASO rainfall, for 31 years 2070–2100 compared to 2001–2031. Figure 3 shows 28 out of 37 model estimates for this region become wetter on average, and most models (i.e. 22 out of 37 models) exhibit increased distribution spreads reflected by raised SDs. Hence many models generally agree on the direction of these changes, but even then the magnitude of changes in GCMs remains uncertain.

Our analysis reveals that current understanding of how future climate change will impact on East Africa ASO drought risk remains uncertain. This is based on a relatively simple assessment of 37 climate models, each given equal weight but after being corrected by observation-based rainfall products. Strong sources of uncertainty in drought prediction include the following: (1) the choice of bias correction methodology; (2) the choice of observational product used to correct bias in GCMs; and (3) the choice of GCMs used. Currently, for many geographical regions, GCM estimates of rainfall changes varies substantially across models (Knutti and Sedláček, 2013). Multi-model analyses such as ours therefore illustrate uncertainty associated with different model parameterisation or scheme describing rainfall features. However, to give more definitive answers, the climate research community may need to be confident enough to rank climate models based on performance to refine future projec-

tions (Knutti et al., 2017). Improving GCM projections will most likely need ongoing constraint of many model components. For East African rainfall predictions in particular, this needs to link to accurate forward projections of oceanic variability. Strong teleconnections are known to exist between El Niño–Southern Oscillation (ENSO) and East African rainfall (Segele et al., 2009; Gissila et al., 2004; Gleixner et al., 2016), and with longer-term fluctuations in Pacific sea surface temperatures, either increasing or decreasing rainfall (Funk et al., 2014; Liebmann et al., 2014; Gleixner et al., 2017 TS5). Larger ensembles of simulations by each model are also important, and especially when analysing the probability of extreme events. This enables a more complete sampling of probability distributions, describing more fully the internal variability of the climate system imposed over general climate changes. Some GCMs estimate an increase in future variability of East African ASO rainfall, and better knowledge of the magnitude of this is important. Significantly raised variability may cause a higher frequency of droughts, even if background trends are for higher mean rainfall levels. Other researchers also illustrate that any variability increases as well as mean changes have strong impacts on society (Brown and Lall, 2006). Furthermore, food and water availability in East Africa has multiple socio-economic drivers, alongside climatic influences (Little et al., 2001; Adhikari et al., 2015). Although here we have focused on climate model projections of the future, more holistic approaches will combine climate and crop impact modelling. The hope is that climate model predictions for East Africa will move towards a consensus on expected changes, therefore facilitating better protection and disaster preparedness against future famine.

*Data availability.* . TS6

**The Supplement related to this article is available online at https://doi.org/10.5194/nhess-18-1-2018-supplement.**

*Competing interests.* The authors declare that they have no conflict of interest. TS7

*Acknowledgements.* Hui Yang gratefully acknowledges funding from the China Scholarship Council, and Chris Huntingford acknowledges the NERC CEH National Capability fund. The authors acknowledge the World Climate Research Programme's Working Group on Coupled Modelling, which is responsible for CMIP, and we thank the climate modelling groups for producing and making available their model output. We also acknowledge the re-analysis products of the European Centre for Medium-Range Weather Forecasts.

Edited by: Bruno Merz
Reviewed by: two anonymous referees

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

**Remarks from the language copy-editor**

CE1    Should this be $1\sigma$ (or $2\sigma$) throughout?

**Remarks from the typesetter**

TS1    The composition of all figures has been adjusted to our standards.

TS2    Copernicus Publications collects the DOIs of data sets, videos, samples, model code, and other supplementary/underlying material or resources as well as additional outputs. These assets should be added to the reference list (author(s), title, DOI, and year) and properly cited in the article. If no DOI can be registered, assets can be linked through persistent URLs. This is not seen as best practice and the persistence of the URL must be secured.

TS3    Please check if any section structure should be added to the paper e.g. introduction, conclusions.

TS4    Please add reference to the reference list.

TS5    Please add reference to reference list.

TS6    Please provide a statement on how your underlying research data can be accessed. If the data are not publicly accessible, a detailed explanation of why this is the case is required. The best way to provide access to data is by depositing them (as well as related metadata) in reliable public data repositories, assigning digital object identifiers (DOIs), and properly citing data sets as individual contributions. Please indicate if different data sets are deposited in different repositories or if data from a third party were used. If no DOI is available, assets can be linked through persistent URLs to the data set itself (not to the repositories' home page). This is not seen as best practice and the persistence of the URL must be secured.

TS7    Declaration of all potential conflicts of interest is required by us as this is an integral aspect of a transparent record of scientific work. If there are possible conflicts of interest (see http://publications.copernicus.org/services/competing_interests_policy.html), please state what competing interests are relevant to your work.

TS8    Please provide last access date.

TS9    Reference not mentioned in text.

TS10    Please provide article number with DOI or page range.

TS11    Please provide volume with article number and DOI or page range.

TS12    Please provide more information.

TS13    Reference not mentionted in text.