# Peer review of "Brief Communication: Drought Likelihood for East Africa"

_Natural Hazards and Earth System Sciences, 2017_

## Referee Comment (RC1) · Anonymous Referee #1 · 6 Jun 2017

**1. SUMMARY**

The manuscript discusses the ongoing drought in East Africa and asks the question if there is a link between the drought and increasing greenhouse gas levels. The study uses both ERAinterim reanalysis data and CMIP5 models in order to try to establish this link.

**2. GENERAL REMARKS**

METHOD: The method is rather experimental and not well documented or explained. Even within the tight limits of this manuscript type, there would be plenty of room to explain the method and if it can indeed explain what is going on.

FIGURES: The figures are readable and do not have to be re-done. However, from

the figures, rather than focusing on the very small changes for the current level of precipitation, it would be worthwhile focusing on the clear change in the overall distribution, showing a significant increase in inter-annual variability, which is strongly linked to socio-economic indicators (see e.g. Brown & Lall, 2006).

REFERENCES: It is very surprising that no climate study is cited that looks at the impact of climate change on East African rainfall instead of just impact studies. Also, the impact of external forcings such as ENSO is not mentioned, although for East Africa that might be a major factor for the strong (and increasing) inter-annual variability.

LANGUAGE: The English should be improved.

CONCLUSION: The conclusion offers a smorgasbord of other studies, and does not help the reader understand what the present study is able to contribute to the current research. Instead, it suggests that other methods may be more worthwhile exploring. It will have to be made much more clear what the benefit of this study is in order for it to be published.

3. DETAILED COMMENTS

Page 1:

Line 10: "merging" could be explained better

Line 11: GCM is the acronym for "General Circulation Model"

Line 11: make sure to distinguish ERAinterim from other ECMWF reanalyses

Line 18: the reader would need at least some justification why there was a famine in East Africa and not in other regions, where according to Fig. 1a the rainfall deficiency is much worse

Fig. 1: that does not look like the percentage of average rainfall, as suggested in the figure, rather it must be the percentage deviation from the mean rainfall.

NHESSD
Line 18: "this" = this region?

Line 19: a list of the models would be helpful. Readers like to know which model they're looking at, and if these particular models were picked for a reason.

Lines 20 – 23: describe the method more clearly.

Line 22: "This is also...": this sentence is not clear.

Line 22: precipitation estimates in reanalysis products tend to be comparably poor. It will need to be justified why this particular dataset was used and not some other precipitation dataset.

Line 24: "31 times 37 numbers": be more clear

Line 25: month-1 = per month?

Line 27/28: is this a significant increase? It seems rather small.

Line 28: "stretch in the distribution tail"? maybe just describe that the mean of the distribution shifts to higher rainfall amounts, while the tails flatten.

Line 29: "stretched left-tails": same here

Line 30: "a few models": which ones? How many?

Line 30: "increased interannual variability": it would be helpful for the general reader to know the seasonal cycle of rainfall in this region. It seems there is a significant intermodel variability, and it does not become clear from the manuscript if these models can be trusted.

Line 1: "considers models equally": but the models are all modified to fit ERAinterim, so "equally" is maybe not the right term?

Lines 2 - 7: this conclusion has to be improved. Some sentences suggest other ap-
proaches may be better suited to study this problem, while some bring up topics that should have been covered in the introduction. What is the conclusion from your own method?

Figure 1b: is this year significantly different from other years? what about these other years when rainfall was low or even lower than this year? Were these also drought years?

Figure 1c: the PDFs look surprisingly smooth, it would have been nicer to see some structure. Or at the very least explain the smoothing that has been used.

Cited Reference: Brown, C. and Lall, U. (2006), Water and economic development: The role of variability and a framework for resilience. Natural Resources Forum, 30: 306–317. doi:10.1111/j.1477-8947.2006.00118.x

---

## Referee Comment (RC2) · Anonymous Referee #2 · 20 Jun 2017

**Review for NHESS submission, "Brief Communication: Drought Likelihood for East Africa" by Yang and Huntingford**

**General comments**

This study by Yang and Huntingford uses a combination of historical data from a single reanalysis product (ERA-Interim), combined with models from the CMIP5 suite, to quantify changes in drought risk over the East Africa region under a high-warming scenario (RCP8.5). The study does not include validation analysis to determine whether the models being considered can in fact provide a reasonable representation of the observed climate for the region of interest – this is particularly problematic given the long-recognised absence of high-quality observational records over these regions. There have also been several other studies analysing changes to East African drought in recent years, which the authors have failed to acknowledge. Finally, there is little to no treatment of statistical uncertainty estimates accompanying the results, which is fundamentally misleading to the reader.

Therefore, my recommendation is that *this article should be rejected*. In accordance with the NHESS review criteria, my assessment is as follows:

Scientific Significance: 4 (poor)

Scientific Quality: 4 (poor)

**Presentation Quality: 3 (fair)**

\_\_\_\_\_

**Specific comments**

**1)** Why was ERA-Interim reanalysis chosen? Has there been any sensitivity analyses using other reanalysis products, or comparison with observational rainfall products for common time periods? I would like to see some demonstration as to whether the results would vary if NCEP2, JRA or 20th Century Reanalysis products were used instead, as well as some consideration of actual observational data (such as TRMM, CRU-TS, or individual station-based records).

The authors are referred to the following papers for reference:

- Angelil et al (2016, Weather and Climate Extremes, https://doi.org/10.1016/j.wace.2016.07.001)
- Sillmann et al (2013, JGR Atmospheres, doi:10.1002/jgrd.50203)

**2)** Why is there no broader consideration of the suitability of the climate models used, beyond a bias correction based on climatological mean precipitation? This approach to bias correction has been widely considered inadequate in the context of analysing precipitation extremes, and especially so for within an attribution context. For example, if the width of the distribution of precipitation is unrealistically narrow, than even a 'realistic' shift in the mean of the distribution would result in an overestimation of the increased likelihood of future low-precipitation extremes.

The authors are referred to the following papers for reference:

- Sippel et al (2016, Earth System Dynamics, doi:10.5194/esd-7-71-2016)
- Jeon et al (2016, Weather and Climate Extremes, <a href="https://doi.org/10.1016/j.wace.2016.02.001">https://doi.org/10.1016/j.wace.2016.02.001</a>)
- Angelil et al (2017, Journal of Climate, https://doi.org/10.1175/JCLI-D-16-0077.1)

**3)** The authors don't mention the wide variety of papers already in the peer-review literature that have evaluated changes to East African drought under a warming climate, and in my opinion, this manuscript does not add sufficient value beyond these previous studies to warrant publication in a high-quality journal like *NHESS*. Moreover, the authors fail to consider the many other contributions beyond low precipitation which

can contribute to a severe drought. In this context, an exploration of more robust drought metrics would be helpful.

The most relevant previous studies for the authors include:

- Lott et al (2013, *Geophysical Research Letters*, 10.1002/grl.50235)
- Lott et al (2014, Environmental Research Letters, https://doi.org/10.1088/1748-9326/9/10/104017)
- Funk et al (2015, Bulletin of the American Meteorological Society, DOI: 10.1175/BAMS-D-15-00106.1)
- Marthews et al (2015, *Bulletin of the American Meteorological Society*, DOI:10.1175/BAMS-D-15-00115.1)
- Funk et al (2016, Bulletin of the American Meteorological Society, DOI:10.1175/BAMS-D-16-0167.1)
- Climate Central (2017, https://wwa.climatecentral.org/analyses/kenya-drought-2016/)

**4)** The absence of uncertainty estimates in Figure 1c is troubling. Most attribution studies provide, at the very least, bootstrapping estimates of uncertainty. I suspect that if error bars did accompany the probability changes in the inset panel, the changes for 2001-2031 and 2035-2065 would be statistically insignificant relative to 1861-1891. Further, it is not clear as to whether the future probabilities of witnessing less than 46mm month-1 have been calculated using raw model data, or the excessively-smoothed PDF constructions.

**5)** The use of 1861-1891 is a misleading representation of 'pre-industrial' as it includes the influences of the Krakatoa volcanic eruption, and associated cooling effects. I strongly recommend changing the baseline period, perhaps to 1861-1880 in accordance with the IPCC's definition.

\_\_\_\_\_

**Technical Comments:**

**Specific comments:**

P1, L9: Are these small increases statistically significant?

P1, L10: East Africa, not East African.

P1, L10: It's not really an analysis 'merging' ERA-Interim data with CMIP5 data. Instead you are using ERA-Interim as a basis for bias-correcting the model data.

P1, L14: '... shows that during August to October'?

P1, L20: RCP8.5 is not designed as a 'business-as-usual' scenario. It is in fact an acceleration of the presentday rates of increase in radiative forcing. The closest to 'business-as-usual' would be RCP6.0. I suggest the authors modify this particular phrase.

P1, L20-L23: This is poorly phrased. So you are bias-correcting based on climatological mean precipitation for ERA-Interim?

P1, L27-28: I suspect a change from 5.6 to 5.8% is not statistically significant in any way, and repeating these calculations with even just two or three models removed could lead to a completely different answer. What is the range in answers of this probabilistic increase for each individual model?

P1, L29: 'stretched left-tails' is a poor description, and I suggest changing this.

P2, L2-7: This is a very poor concluding paragraph. You do not summarise the key results of the study, but instead offer reasons why significance testing is needed (reliability of different models), before highlighting all of the reasons why monthly-mean precipitation deficits may in fact be a poor proxy for drought impacts over East Africa.

---

## Author Comment (AC1) · 26 Jun 2017

Please see below for corrected Figure 1c. We will reply to the full set of reviewers' comments soon.

[Figure]

We are grateful for the received reviews. We recognise they are very critical, but we will try and answer the points over the next couple of weeks.

The purpose of this immediate response is that the reviewers' questioning of appropriateness of smoothing in our original panel 1c has revealed a factual error. We want to correct that as fast as possible, and alert anyone downloading the current discussion version.

Below are the normalised GCM estimates of mean August-to-October (ASO) rainfall, for different 31-year periods and for 37 GCMs. This is for same study region (black rectangle, original Figure 1a). A normal distribution was fitted, which "by eye" performs reasonably well for 1861-1891 and 2001-2031. However in the histogram for later periods, a skewness appears. The tail on the right-hand side becomes longer and higher. This is caused by some very high predicted future rainfall values (i.e. > 100 mm month$^{-1}$). Due to the symmetry implicit in the normal distribution, this artificially enhances the estimated probability of drought occurrence. That is, the fitted normal distributions have left-hand lower tails that are too large and too high, when compared to GCM estimates.

If instead we calculate the probability of drought occurrence using directly the normalised GCMs values - so from the histogram itself - we get a revised plot (right-hand plot, below). This implies a decreasing future risk of severe drought, based on the ASO 2016 rainfall threshold for our East Africa region.

In any revised document, we will not use fitted curves, and instead present the histograms directly.

[Figure]

[Figure]

**Fig. 1.**

---

## Author Comment (AC6) · 22 Aug 2017

For our responses to reviewer 2, please see the uploaded and combined replies under reviewer 1 above. Also the manuscript (with and without track changes) is presented under our response to reviewer 1.

---

## Author Response (AR1)

Dear Editor of NHESS

Thank you for your help in obtaining two reviews of our Brief Communication:

**Drought Likelihood for East Africa.**

The drought in East Africa has caused severe problems for millions of people, and including high numbers of deaths. Although there are plots of climate model projections of rainfall changes in the IPCC report, these are not specific to East Africa. Instead what we are trying to achieve in this short manuscript is a straightforward "show and tell" of future model projections of rainfall for this region, and for specific period August, September and October.

The research community needs to obtain a full meteorological understanding of what happened in year 2016, and then apply this knowledge to assess its occurrence in future projections by climate models. We hope a set of standard comprehensive climate research papers on African drought will appear over the years ahead. Our brief communication may act to encourage this, along with highlighting the need to refine future estimates of change.

We recognize our reviews are poor, and especially from Reviewer 2. We have, though, responded in full to all requests. This includes making all methods transparent, and in particular providing uncertainty bounds on (a) rainfall datasets and (b) bias correction methodology to scale climate model to measurement records. We believe our paper now provides a preliminary but robust assessment of Africa drought risk.

We hope that your journal will consider our responses, and in light of these that there might still be the possibility of publishing our analysis as a Brief Communications in NHESS. Our responses are described below. Adjusted manuscript text is repeated in red font. We believe we have satisfied all formatting requirements.

Thank you for all the help so far.

With kind regards,

Hui Yang (and on behalf of Chris Huntingford).

**Response to the reviewers**

**To Reviewer #1:**

*Reviewer #1 General Comments:*

We thank Reviewer #1 for their help and time spent on this paper. Our responses are below, and revised text in the paper repeated in red font.

**METHOD**: *The method is rather experimental and not well documented or explained. Even within the tight limits of this manuscript type, there would be plenty of room to explain the method and if it can indeed explain what is going on.*

**[Response 1.1]** We accept there is a need for a more complete description of the calculation methods used, which we have now done. The main change is that in the revised manuscript we now write: "A bias correction with two post-processing steps is applied to the GCM precipitation estimates. We first calculate modelled and ERA-based mean ASO rainfall estimates over the east Africa during the period 1979-2015. The GCM precipitation estimates, both past and future, are corrected by a GCM-specific mean correction factor, which is a ratio of the climatological mean of each GCM to that of the ERA-interim reanalysis product. Second, we then adjust the climatological standard deviation (STD) of GCM precipitation estimates by multiplying the ratio of the climatological STD of each GCM to that of the ERA-interim data. The adjustment of spread of rainfall distribution is an important additional procedure to further constrain GCM estimates (Sippel et al., 2016; Jeon et al., 2016; Angelil et al., 2017). Together this ensures all GCMs have the ERA-based mean and STD for period 1979-2015." (Page: 1, Lines: 28-31 and Page: 2, Lines: 1-4)

**FIGURES:** *The figures are readable and do not have to be re-done. However, from the figures, rather than focusing on the very small changes for the current level of precipitation, it would be worthwhile focusing on the clear change in the overall distribution, showing a significant increase in inter-annual variability, which is strongly linked to socio-economic indicators (see e.g. Brown & Lall, 2006).*

**[Response 1.2]** We now added a new figure (i.e. Figure 3), which places an emphasis on GCM-specific changes in mean and standard deviation of ASO rainfall (See Response 1.17, where Figure 3 is shown). We also cite your suggested reference – thank you for alerting us to this paper. We quantify throughout variability by standard deviation (STD), recognizing that any auto-correlation means that strictly speaking, is not its inter-annual variability. We

clarify this in the modified revised manuscript.

*REFERENCES: It is very surprising that no climate study is cited that looks at the impact of climate change on East African rainfall instead of just impact studies. Also, the impact of external forcings such as ENSO is not mentioned, although for East Africa that might be a major factor for the strong (and increasing) inter-annual variability.*

[Response 1.3] We thank the reviewer for this comment, and in the meantime have identified references that we should have used in the initial submission. Using key reference, we now note the potential driving factors of the East African rainfall deficits, and in particular the impact of ENSO. Please see our paper amendments, as follows: "Improving GCM projections also could involve on-going constraining of model components. For rainfall of east Africa predictions in particular, this will link to accurate forward projections of oceanic variability. Strong teleconnections are known to exist between El Niño Southern Oscillation (ENSO) and East African rainfall (Segele et al., 2009; Gissila et al., 2004), and with longer-term fluctuations in Pacific SSTs increasing/decreasing rainfall (Funk et al., 2014; Liebmann et al., 2014)." (Page: 3, Lines: 19-23)

*LANGUAGE: The English should be improved.*

[Response 1.4] We have polished the language in large parts of manuscript, and in addition the technical details are given in a more precise way. We hope the new paper version has a level of clarity as to be useful to a relatively broad audience.

*CONCLUSION: The conclusion offers a smorgasbord of other studies, and does not help the reader understand what the present study is able to contribute to the current research. Instead, it suggests that other methods may be more worthwhile exploring. It will have to be made much more clear what the benefit of this study is in order for it to be published.*

*Page 2, Lines 2 – 7: this conclusion has to be improved. Some sentences suggest other approaches may be better suited to study this problem, while some bring up topics that should have been covered in the introduction. What is the conclusion from your own method?*

[Response] We make clearer our contribution, which is to find we cannot yet make accurate projections of change, as this depends on GCM, any bias-correction method used, and the observation product used in such bias removal. Although this is a slightly negative find (i.e. answer is inclusive), we believe this is still important to have placed in the literature. We are targeting "Brief Communication" format, so the issues brought up are not expanded in full,

but we would be grateful to mention in the context of our analysis.

The many issues raised are dealt with throughout the concluding paragraphs, and although it takes up space in our response, we list this in full as: "Our analysis reveals that current understanding of how future climate change will impact on East Africa ASO drought risk remains uncertain. This is based on a relatively simple assessment of 37 climate models, each given equal weight but after being corrected by observation-based rainfall products. We find the sources of uncertainty in drought prediction include: 1) the choice of bias correction methodology; 2) the choice of observational product used to correct bias in GCMs; and 3) the choice of GCMs used. Currently, for many geographical regions, GCM estimates of rainfall changes varies substantially across models (Knutti and Sedláček, 2013). Multi-model analyses such as ours consider uncertainty associated with different model parameterisation or scheme describing rainfall features. However, to give more definitive answers, the climate research community may need to be confident enough to rank climate models based on performance to refine future projections (Knutti et al., 2017). Improving GCM projections also could involve on-going constraining of model components. For rainfall of east Africa predictions in particular, this will link to accurate forward projections of oceanic variability. Strong teleconnections are known to exist between El Niño Southern Oscillation (ENSO) and East African rainfall (Segele et al., 2009; Gissila et al., 2004), and with longer-term fluctuations in Pacific SSTs increasing/decreasing rainfall (Funk et al., 2014; Liebmann et al., 2014). Larger ensembles of simulations by each model is also important, and especially when analysing the probability of extreme events. This enables a more complete sampling of probability distributions, describing more fully the internal variability of the climate system imposed over general climate change. In addition, some GCMs estimate an increase in future variability of east African ASO rainfall, and better knowledge of the magnitude of this is important. Research shows any variability increases as well as mean changes has strong impacts on society (Brown and Lall, 2006). Furthermore, food and water availability in East Africa has multiple socio-economic drivers, alongside climatic influences (Little et al., 2001; Adhikari et al., 2015). Although here we have focused on climate model projections of the future, more holistic approaches will combine climate and crop impact modelling. The hope is that climate model predictions for east Africa will move towards a consensus on expected changes, helping then better protection and disaster preparedness against future famine." (Page: 3, Lines: 12-33)

*Reviewer #1* **Detailed Comments:**

*Page 1, Line 10: "merging" could be explained better*

**[Response 1.5]** We have clarified this in the manuscript: "After bias correction to match contemporary rainfall mean, GCMs project small decreases in probability of drought of same severity for East Africa by the end of 21st century. However, further adjusting the variance of GCMs to match ERA-interim data, probability of drought increases slightly." (Page: 1, Lines: 9-11)

*Page 1, Line 11: GCM is the acronym for "General Circulation Model"*

**[Response 1.6]** Corrected.

*Page 1, Line 11: make sure to distinguish ERAinterim from other ECMWF reanalyses*

**[Response 1.7]** Done.

*Page 1, Line 18: the reader would need at least some justification why there was a famine in East Africa and not in other regions, where according to Fig. 1a the rainfall deficiency is much worse*

**[Response 1.8]** The east Africa is especially vulnerable to the impacts of drought because of a unique combination of several adverse factors. We now write in paper: "We concentrate on East Africa, as this region experienced particularly poor harvest and where famine was widely reported during 2016 (noting that regions outside black rectangle of Fig. 1a also experienced major rainfall deficits in 2016). East Africa is especially vulnerable to the impacts of drought (DEC, 2017). The region has long experienced widespread poverty and high levels of food insecurity (Von Grebmer et al., 2016). The high dependence of its population on rain-fed agriculture, sometimes in tandem with political instability, exacerbate the impacts of droughts (Love, 2009; Masih et al., 2014)." (Page: 1; Lines: 18-24).

*Fig. 1: that does not look like the percentage of average rainfall, as suggested in the figure, rather it must be the percentage deviation from the mean rainfall.*

**[Response 1.9]** Thank you for this. The Figure label now reads "Rainfall for Aug to Oct, 2016 as a percent change from the average (%)".

*Page 1, Line 18: "this" = this region?*

**[Response 1.10]** Done.

*Page 1, Line 19: a list of the models would be helpful. Readers like to know which model they're looking at, and if these particular models were picked for a reason.*

**[Response 1.11]** We list the CMIP5 ESMs that provide monthly precipitation of both historical simulations and RCP8.5 projections in this study, resulting in a list of 37 ESMs (Table S1). In addition, our new Figure 3 provides GCM-specific information on projections (see Response 1.17 for detailed explanations).

**Table S1**. CMIP5 global circulation models (GCMs) used in this study, and their components.

| Model Name | Atmospheric Model | Land surface Model | Oceanic Model | Reference |
|---|---|---|---|---|
| ACCESS1-0 | HadGEM2 r1.1 | MOSES | MOM4pl | *Bi et al.*(2012) |
| ACCESS1-3 | Similar to GA 1.0 | CABLE v1.8 | MOM4p | |
| bcc-csm1-1 | BCC_AGCM2.2 | BCC_AVIM1.0 | MOM4_L40 | *Wu et al.* (2013) |
| bcc-csm1-1-m | BCC_AGCM2.2 | BCC_AVIM1.0 | MOM4_L40 | |
| BNU-ESM | CAM3.5 | CLM | MOM4p1 | *Ji et al.* (2014) |
| CanESM2 | CanAM4 | CLASS2.7 | CanOM4 and CMOC1.2 | *Arora et al.* (2011) |
| CCSM4 | CAM4 | CLM4 | POP2 | *Gent et al.* (2011) |
| CESM1-BGC | CAM4 | CLM4 | POP2 | *Neale et al.* (2010) |
| CESM1-CAM5 | CAM5 | CLM4 | POP2 | |
| CMCC-CESM | ECHAM5 | SILVA | NEMO | *Scoccimarro et al.* (2011) |
| CMCC-CM | ECHAM5 | SILVA | OPA 8.2 | |
| CMCC-CMS | ECHAM5 | SILVA | OPA 8.2 | |
| CNRM-CM5 | ARPEGE climate | SURPEXv5.1 | NEMO3.3 | *Voldoire et al.* (2011) |
| CSIRO-Mk3-6-0 | AGCMv7.3.8 | a soil-canopy scheme | GFDL MOM2.2 | *Rotstayn et al.* (2010) |
| EC-EARTH | IFS | H-TESSEL | NEMO | Hazeleger et al. (2010) |
| GFDL-CM3 | GFDL-AM3 | LM3 | MOM | *Donner et al.* (2011) |
| GFDL-ESM2G | GFDL-AM2.1 | LM3 | GOLD | *Dunne et al.* (2012) |
| GFDL-ESM2M | GFDL-AM2.1 | LM3 | MOM4 | |
| GISS-E2-H-CC | GISS-E2 | GISS-LSM-CC | HYCOM | *Schmidt et al.* (2014) |
| GISS-E2-H | GISS-E2 | GISS-LSM | HYCOM | |
| GISS-E2-R-CC | GISS-E2 | GISS-LSM-CC | Russell | |
| GISS-E2-R | GISS-E2 | GISS-LSM | Russell | |
| HadGEM2-CC | HadGAM2 | TRIFFID | HadGOM2 | *Collins et al.* (2011) |
| HadGEM2-ES | HadGAM2 | TRIFFID | HadGOM2 | *Jones et al.* (2011) |
| INMCM4 | INM | INM | HadGOM2 | *Volodin et al.* (2010) |
| IPSL-CM5A-LR | LMDZ5A | ORCHIDEE | NEMO | *Dufresne et al.* (2012) |
| IPSL-CM5A-MR | LMDZ5A | ORCHIDEE | NEMO | |
| IPSL-CM5B-LR | LMDZ5B | ORCHIDEE | NEMO | |
| MIROC5 | FRCGC-AGCM | MATSIRO | COCO4.5 | *Watanabe et al.* (2011) |
| MIROC-ESM | FRCGC-AGCM | MATSIRO | COCO4.5 | |
| MIROC-ESM-CHEM | FRCGC-AGCM | MATSIRO | COCO4.5 | |
| MPI-ESM-LR | ECHAM6 | JSBACH | MPIOM | *Ilyina et al.* (2013) |
| MPI-ESM-MR | ECHAM6 | JSBACH | MPIOM | |
| MRI-CGCM3 | MRIÒAGCM3 | HAL | MRI.COM3 | *Yukimoto et al.* (2012) |
| NorESM1-ME | CAM4-Oslo | CLM4 | MICOM | *Tjiputra et al.* (2013) |
| NorESM1-M | CAM4-Oslo | CLM4 | MICOM | |

*Page 1, Lines 20 – 23: describe the method more clearly.*

*Page 1, Line 22: "This is also...": this sentence is not clear.*

**[Response]** Yes, the original manuscript version we now see could have been written more clearly. We now re-write the method in the revised manuscript as follows, and we think it now has ambiguity removed: "A bias correction with two post-processing steps is applied to the GCM precipitation estimates. We first calculate modelled and ERA-based mean ASO rainfall estimates over the east Africa during the period 1979-2015. The GCM precipitation estimates, both past and future, are corrected by a GCM-specific mean correction factor, which is a ratio of the climatological mean of each GCM to that of the ERA-interim reanalysis product. Second, we then adjust the climatological standard deviation (STD) of GCM precipitation estimates by multiplying the ratio of the climatological STD of each GCM to that of the ERA-interim data. The adjustment of spread of rainfall distribution is an important additional procedure to further constrain GCM estimates (Sippel et al., 2016; Jeon et al., 2016; Angelil et al., 2017). Together this ensures all GCMs have the ERA-based mean and STD for period 1979-2015." (Page: 1, Lines: 28-31 and Page: 2, Lines: 1-4)

*Page 1, Line 22: precipitation estimates in reanalysis products tend to be comparably poor. It will need to be justified why this particular dataset was used and not some other precipitation dataset.*

**[Response 1.12]** This is also noted by reviewer 2, and we answer this in full by now using a large ensemble of four additional global precipitation data sets (Table S2). These precipitation data sets are interpolated gauge observations only (i.e. PREC/L and CPC), gauge observations combined with satellite measurements (i.e. GPCP), or satellite observations (i.e. TRMM). This is alongside our original ERA-interim data product.

All the rainfall products show the August-to-October (ASO) rainfall in year of 2016 is less than the climatological mean of ASO rainfall (Table S2 and an additional Figure below which is not in paper). However, when we use all five products to correct climatological mean and STD of GCM rainfall estimates, such bias-correction influences strongly future projections. We illustrate these differences in a new Figure 2, which has one panel of future projections for each rainfall product. We are grateful for this reviewer request, now allowing better presentation of uncertainty.

At lines 21-34 (Page 2) of the paper, we now write: "Given that large uncertainty in the observation-based precipitation products has been well reported (Angélil et al., 2016), we use four other precipitation estimates (GPCP, PREC/L, CPC and TRMM) to bias-correct

GCM estimates. In Fig. 2 we reproduce the insets of Fig 1c (no hatching) and Fig 1d (hatching) for ERA-Interim, and then for the four other precipitation products. Consistent with the conclusions based on the ERA-interim product only, the results from the other rainfall products also show that the probability of drought occurrence in the east Africa has decreased slightly from pre-industrial to present day, and irrespective of whether variance adjustment has occurred (Fig. 2, all blue and black bars, with and without hatching). Future projections, though, of drought likelihood do vary across different precipitation products. For the mean-corrected GCM estimates, 4 out of 5 rainfall product-corrected GCM projections give a slight decrease in drought occurrence likelihoods by the end of 21st century. The exception is the TRMM-corrected GCMs, which suggest the drought probability would increase slightly by 2070-2100 and relative to the present day. For the mean- and variance-corrected GCM estimates, relative to the present-day levels the GCM estimates corrected to the ERA-interim, GPCP, and TRMM products give an increase in drought occurrence probability. However PREC/L- and CPC-corrected GCM estimates suggest the probability of drought occurrence will decrease. This divergence is due to the strong differences in the climatological mean, standard deviation and year 2016 ASO rainfall levels among the different precipitation products (Table S2)."

**Table S2**. The mean August-to-October (ASO) rainfall (mm month$^{-1}$) of year 2016, multi-year mean (not including 2016) and multi-year standard deviation (STD) over east Africa for years 1979 to 2016. The five global precipitation data sets used are listed. Four products of ERA-interim, GPCP, PREC/L, CPC and TRMM are available from 1979 to 2016. These four precipitation data sets are either interpolated gauge observations only (i.e. PREC/L and CPC), gauge observations combined with satellite measurements (i.e. GPCP), or reanalysis data (i.e. ERA-interim). The TRMM satellite observations are available from 2001 to 2016.

| ASO rainfall (mm month$^{-1}$) | ERA-interim | GPCP | PREC/L | CPC | TRMM |
|---|---|---|---|---|---|
| 2016 | 46.10 | 46.56 | 57.16 | 35.78 | 32.05 |
| Climatological mean (1979-2015) | 70.76 | 62.78 | 61.68 | 43.44 | 60.69* |
| Climatological STD (1979-2015) | 11.28 | 10.40 | 11.48 | 13.89 | 11.83* |

* TRMM satellite precipitation data is only available from 2001 to 2016. The climatological ASO rainfall averages of the period 2001-2015 is computed.

[Figure]

**Additional Figure for response (not in paper).** The monthly total rainfall (mm per month) over study region (panel Figure 1a; land within black rectangle) for years 1979 to 2016. Year 2016 is red, other years are individual grey lines, and multi-year average (not including 2016) is blue. Blue shading is ± one standard deviation of monthly rainfall across years 1979-2015. The drought event (shaded in yellow) is defined as the three consecutive months of ASO when rainfall in year 2016 is below blue shading.

[Figure]

**New Figure 2 in the revised manuscript**. CMIP5-based histograms of probabilities of mean ASO rainfall falling below year 2016-based threshold values. Shown for periods 1861-1891 (blue), 2001-2031 (black), 2035-2065 (orange) and 2070-2100 (red). Each bar corresponds to merged normalized outputs from 37 CMIP5 models forced by historical emissions and RCP8.5 future scenario. The bars without horizontal hatching (left) are for the mean-corrected GCM precipitation estimates. The bars with hatching (right) are for the mean- and variance-corrected GCM estimates.

*Page 1, Line 24: "31 times 37 numbers": be more clear*

**[Response 1.13]** We have improved all the related text to number of models and time-periods used. We now write "Bias-corrected mean ASO rainfall are presented in Fig. 1c for mean bias correction, and in Fig. 1d for mean and STD bias correction. These are derived from 37 GCMs, and for four 31-year periods (pre-industrial, present day, and two future periods)." (Page: 2, Lines: 5-7)

*Page 1, Line 25: month-1 = per month?*

**[Response 1.14]** Corrected.

*Page 1, Line 27/28: is this a significant increase? It seems rather small.*

**[Response 1.15]** Based on this, and to enable better assessment of whether these changes are

significant, we now provide uncertainty bounds on probability of drought for different time periods. This allows far better visual comparisons of the changes within Figure 1 and new Figure 2. We do this via bootstrapping methods. In particular, the one standard deviations are estimated via bootstrapping with 80% replications from the 37 GCM precipitation data and for the 31-year periods.

*Page 1, Line 28: "stretch in the distribution tail"? Maybe just describe that the mean of the distribution shifts to higher rainfall amounts, while the tails flatten.*

*Page 1, Line 29: "stretched left-tails": same here*

**[Response 1.16]** As can be seen from our revised Figure 1, we no longer fit normal curves, and instead rely on direct presentation of the probability density functions of ASO rainfall estimates. Hence we have removed the sentences about the changes in the distribution tails in the reviewed manuscript.

*Page 1, Line 30: "a few models": which ones? How many?*

*Page 1, Line 30: "increased interannual variability": it would be helpful for the general reader to know the seasonal cycle of rainfall in this region. It seems there is a significant intermodal variability, and it does not become clear from the manuscript if these models can be trusted.*

**[Response 1.17]** This request has led to new Figure 3 (repeated below), that shows individual model responses. Overall ASO mean rainfall levels increase in 28 out of 37 GCMs from the present period 2001-2031 to the period 2070-2100.

In terms of variability, this new figure in the revised manuscript shows changes in STDs for each GCM. We found enhanced STDs of 31-year rainfall variations in 22 out of 37 GCMs. We have taken the issue of whether the STD of models can be trusted by now including this in our bias-correction methods. Unlike the previous manuscript version, we now compensate for both the model mean and STD discrepancies (Figure 1c versus new Figure 1d, and within Figure 2).

Besides, we write in the text: "We also show individual model changes in mean and STD of ASO rainfall, for 31 years 2070-2100 compared to 2001-2031. Fig. 3 shows 28 out of 37 model estimates for this region become wetter, and most models (i.e. 22 out of 37 models) exhibiting increased distribution spreads reflected by raised STDs." (Page: 3, Lines: 6-9)

[Figure]

**New Figure 3 in the revised manuscript**. Changes in drought frequency, multi-year mean and standard deviations (STD) of 31 consecutive year rainfall amounts. Difference between present period 2001-2031 and period 2070-2100, as estimated by 37 GCMs. GCM estimates are corrected by the ERA-interim rainfall product. Changes to frequencies of drought occurrence are estimated from the mean bias-corrected GCM estimates (1st row), both mean- and variance bias-corrected GCM estimates (2nd row). The colored grids in the 3rd row with black borders indicate statistically significant differences in the 31-year rainfall mean between these two periods (*t*-test, with P < 0.05). The percentage changes are calculated as $[(x_{2070-2100}/x_{2001-2031})-1] \times 100\%$.

*Page 2, Line 1: "considers models equally": but the models are all modified to fit ERAinterim, so "equally" is maybe not the right term?*

**[Response 1.18]** We now modify this sentence, as follow: "This is based on a relatively simple assessment of 37 climate models, each given equal weight but after being corrected by observation-based rainfall products." (Page: 3, Lines: 13-14). Please also note, as described above, our analysis now include five rainfall products, and so not just ERA-interim.

*Figure 1b: is this year significantly different from other years? What about these other years when rainfall was low or even lower than this year? Were these also drought years?*

**[Response 1.19]** The area-weighted spatial average of monthly rainfall from ERA reanalysis product over the August to October (ASO) of 2016 lies at least one standard deviation (STD) below the climatological mean of the other years (i.e. 1979-2015) (see Figure 1b).

There are other five years (1986, 1990, 1991, 1993 and 2010) where the ASO rainfall also lies at least one standard deviation below the climatological mean during 1979-2015.

One year in particular has been studied, that of 2010, and this also caused widespread famine and loss of life. We now write: "For this region, the spatial average of monthly rainfall during ASO of 2016 lies at least one standard deviation below the climatological mean of the other years (Fig. 1b). The year of 2016 is the third driest year in the past four decades. Other years with rainfall at least one standard deviation below the climatological mean during 1979-2015 are 1986, 1990, 1991, 1993 and 2010. Year of 2010 also suffered from the severe famine (Dutra et al., 2013)." (Page: 1; Lines: 15-18)

*Figure 1c: the PDFs look surprisingly smooth, it would have been nicer to see some structure. Or at the very least explain the smoothing that has been used.*

**[Response 1.20]** Please note we responded rapidly to this, as we accept the smoothness is misleading. Please see our earlier response on NHESS website, as to why we reverted to presenting directly the pdf of ASO seasonal rainfall, rather than a smoothed fit to these.

[Figure]

Rainfall for Aug to Oct, 2016 as a percent change from the average (%)

**Figure 1 in the revised manuscript**. (**a**) Black rectangle is location of study region (14.5°N~1.5°S, 36°E~51°E). Plotted is mean rainfall for 2016 and months August to October inclusive (ASO), presented relative changes (as %) to long-term average ASO values (1979-2015). Values based on ERA-interim reanalysis product. (**b**) ERA-based monthly total

rainfall (mm month$^{-1}$) over study region (panel a; land within black rectangle) for years 1979 to 2016. Year 2016 is red, other years are individual grey lines, and multi-year average (not including 2016) is blue line. Blue shading is ± one standard deviation of monthly rainfall across years 1979-2015. The drought event (shaded in yellow) is defined as the three consecutive months of ASO, and when rainfall in year 2016 is below blue shading. (**c**) CMIP5-based PDFs of mean ASO rainfall for periods 1861-1891 (blue), 2001-2031 (black), 2035-2065 (orange) and 2070-2100 (red). Each curve corresponds to the mean-corrected combined outputs from 37 CMIP5 models forced by historical emissions and RCP8.5 future scenario. Individual GCM bias correction is based on the ERA-interim reanalysis product. Yellow shading is mean ASO rainfall less than 46 mm month-1, which is the ERA-interim 2016-based threshold (mean of ASO, red curve in panel b). Inset shows probabilities of mean rainfall of ASO falling below the threshold for the same modelled periods (colours match those of curves). The error bars are the standard deviations (estimated via bootstrapping 80% replications from the 37 GCM precipitation data for the 31-year periods). (**d**) same as (**c**), but based on the mean- and variance-corrected GCM rainfall estimates.

**To Reviewer #2:**

*Reviewer #2* **General Comments:**

We thank Reviewer #2 for their help and time spent on this paper. Our responses are below, and revised text in the paper repeated in red font.

*This study by Yang and Huntingford uses a combination of historical data from a single reanalysis product (ERA-Interim), combined with models from the CMIP5 suite, to quantify changes in drought risk over the East Africa region under a high-warming scenario (RCP8.5). The study does not include validation analysis to determine whether the models being considered can in fact provide a reasonable representation of the observed climate for the region of interest – this is particularly problematic given the long-recognised absence of high-quality observational records over these regions. There have also been several other studies analysing changes to East African drought in recent years, which the authors have failed to acknowledge. Finally, there is little to no treatment of statistical uncertainty estimates accompanying the results, which is fundamentally misleading to the reader.*

**[Response 2.1]** Thank you for these thoughtful and valuable comments on previous version of this manuscript. Following those comments and suggestions, we have thoroughly revised the manuscript. Main changes in the revised manuscript correspond to the three requests above:

1. Given that the large uncertainty in the observation-based precipitation products has been well reported, we have advanced our calculations to include other precipitation products (e.g. GPCP, PREC/L, CPC and TRMM). This allows assessment of the robustness any conclusions.

2. We have cited the recent published papers related to the east African drought.

3. We perform additional analysis, and now apply bootstrapping techniques to provide uncertainty bounds on revised diagrams.

Please see below detailed responses to each specific comment.

*Reviewer #2* **Specific Comments:**

*1) Why was ERA-Interim reanalysis chosen? Has there been any sensitivity analyses using other reanalysis products, or comparison with observational rainfall products for common time periods? I would like to see some demonstration as to whether the results would vary if NCEP2, JRA or 20th Century Reanalysis products were used instead, as well as some consideration of actual observational data (such as TRMM, CRU-TS, or individual station-*

*based records).*

*The authors are referred to the following papers for reference:*

> *- Angelil et al (2016, Weather and Climate Extremes,)*
>
> *- Sillmann et al (2013, JGR Atmospheres, doi:10.1002/jgrd.50203)*

**[Response 2.2]** Based on this request, we now use an ensemble of five global precipitation data sets (Table S2). In addition to our original ERA-interim product, we now use four additional precipitation data sets. These are interpolated gauge observations only (i.e. PREC/L and CPC), gauge observations combined with satellite measurements (i.e. GPCP), or satellite observations (i.e. TRMM).

Although the uncertainty in rainfall changes over Africa is large, all the rainfall products show the August-to-October (ASO) rainfall in year of 2016 is less than the climatological mean ASO rainfall to each (Table S2 and an additional Figure below which is not in paper). However, when using the different datasets to bias-correct the GCM rainfall estimates (their climatological mean and SD), we do find our estimated probabilities of future drought are dependent on precipitation product used. We are grateful for the encouragement to do this, and it has led to a new Figure 2 in the manuscript (please see below).

For completeness, below we repeat here the revised part of the manuscript that describes the differences found in our results from between using five precipitation products to do the bias-correction (and after that follows Table S2, and two Figures mentioned above). At lines 21-34 (Page 2) of the paper, we now write: "Given that large uncertainty in the observation-based precipitation products has been well reported (Angélil et al., 2016), we use four other precipitation estimates (GPCP, PREC/L, CPC and TRMM) to bias-correct GCM estimates. In Fig. 2 we reproduce the insets of Fig 1c (no hatching) and Fig 1d (hatching) for ERA-Interim, and then for the four other precipitation products. Consistent with the conclusions based on the ERA-interim product only, the results from the other rainfall products also show that the probability of drought occurrence in the east Africa has decreased slightly from pre-industrial to present day, and irrespective of whether variance adjustment has occurred (Fig. 2, all blue and black bars, with and without hatching). Future projections, though, of drought likelihood do vary across different precipitation products. For the mean-corrected GCM estimates, 4 out of 5 rainfall product-corrected GCM projections give a slight decrease in drought occurrence likelihoods by the end of 21st century. The exception is the TRMM-corrected GCMs, which suggest the drought probability would increase slightly by 2070-2100 and relative to the present day. For the mean- and variance-corrected GCM estimates, relative to the present-day levels the GCM estimates corrected to the ERA-interim,

GPCP, and TRMM products give an increase in drought occurrence probability. However PREC/L- and CPC-corrected GCM estimates suggest the probability of drought occurrence will decrease. This divergence is due to the strong differences in the climatological mean, standard deviation and year 2016 ASO rainfall levels among the different precipitation products (Table S2)."

**Table S2**. The mean August-to-October (ASO) rainfall (mm month$^{-1}$) of year 2016, multi-year mean (not including 2016) and multi-year standard deviation (STD) over east Africa for years 1979 to 2016. The five global precipitation data sets used are listed. Four products of ERA-interim, GPCP, PREC/L, CPC and TRMM are available from 1979 to 2016. These four precipitation data sets are either interpolated gauge observations only (i.e. PREC/L and CPC), gauge observations combined with satellite measurements (i.e. GPCP), or reanalysis data (i.e. ERA-interim). The TRMM satellite observations are available from 2001 to 2016.

| ASO rainfall (mm month$^{-1}$) | ERA-interim | GPCP | PREC/L | CPC | TRMM |
|---|---|---|---|---|---|
| 2016 | 46.10 | 46.56 | 57.16 | 35.78 | 32.05 |
| Climatological mean (1979-2015) | 70.76 | 62.78 | 61.68 | 43.44 | 60.69* |
| Climatological STD (1979-2015) | 11.28 | 10.40 | 11.48 | 13.89 | 11.83* |

* TRMM satellite precipitation data is only available from 2001 to 2016. The climatological ASO rainfall averages of the period 2001-2015 is computed.

[Figure]

**Additional Figure for response (not in paper).** The monthly total rainfall (mm per month) over study region (panel Figure 1a; land within black rectangle) for years 1979 to 2016. Year 2016 is red, other years are individual grey lines, and multi-year average (not including 2016) is blue. Blue shading is ± one standard deviation of monthly rainfall across years 1979-2015. The drought event (shaded in yellow) is defined as the three consecutive months of ASO when rainfall in year 2016 is below blue shading.

[Figure]

 **New Figure 2 in the revised manuscript**. CMIP5-based histograms of probabilities of mean ASO rainfall falling below year 2016-based threshold values. Shown for periods 1861-1891 (blue), 2001-2031 (black), 2035-2065 (orange) and 2070-2100 (red). Each bar corresponds to merged normalized outputs from 37 CMIP5 models forced by historical emissions and RCP8.5 future scenario. The bars without horizontal hatching (left) are for the mean-corrected GCM precipitation estimates. The bars with hatching (right) are for the mean- and variance-corrected GCM estimates.

*2) Why is there no broader consideration of the suitability of the climate models used, beyond a bias correction based on climatological mean precipitation? This approach to bias correction has been widely considered inadequate in the context of analysing precipitation extremes, and especially so for within an attribution context. For example, if the width of the distribution of precipitation is unrealistically narrow, than even a 'realistic' shift in the mean of the distribution would result in an overestimation of the increased likelihood of future low-precipitation extremes.*

*The authors are referred to the following papers for reference:*

      *-Sippel et al (2016, Earth System Dynamics, doi:10.5194/esd-7-71-2016)*

      *-Jeon et al (2016, Weather and Climate Extremes)*

      *-Angelil et al (2017, Journal of Climate, https://doi.org/10.1175/JCLI-D-16-0077.1)*

**[Response 2.3]** We have performed new analysis in response to this request. In particular, we now additionally bias-correct and ensure the standard deviation of GCMs equals the standard deviation in the observational data products. This has led us to re-calculate the likelihoods of

east African drought, and a new panel in Figure 1 (Figure 1d in the revised manuscript). We are grateful for this request to address distribution width issue - our results suggest the choice of bias correction methodology (i.e. with/without additional STD bias-correction) is a major source of uncertainty in drought likelihood projection.

Then state here we now use those reference, and repeat the sentences here: "Second, we then adjust the climatological standard deviation (STD) of GCM precipitation estimates by multiplying the ratio of the climatological STD of each GCM to that of the ERA-interim data. The adjustment of spread of rainfall distribution is an important additional procedure to further constrain GCM estimates (Sippel et al., 2016; Jeon et al., 2016; Angelil et al., 2017)." (Page: 1; Line: 31 and Page: 2, Lines: 1-3)

*3) Moreover, the authors fail to consider the many other contributions beyond low precipitation which can contribute to a severe drought. In this context, an exploration of more robust drought metrics would be helpful.*

**[Response 2.4]** Thank you for this comment. We respectfully request that we don't expand our paper to alternative drought metrics, in part as some involve socio-economic and governance implications which would take beyond the remit of a Brief Communications. This, hopefully, will be a component of full size climate - socio-economic studies in over the years ahead.

However, we do want to change the manuscript to acknowledge this issue. In particular, other African regions also suffered from rainfall deficits in 2016, and yet there were fewer media reports of famine. This supports, therefore, that drought-induced famine has more aspects than simply low rainfall. For this reason, we want to stress this point, and by guiding readers to observe the low rainfall also recorded beyond East Africa, and as show in Figure 1. We have now amended the manuscript to say: "We concentrate on East Africa, as this region experienced particularly poor harvest and where famine was widely reported during 2016 (noting that regions outside black rectangle of Fig. 1a also experienced major rainfall deficits in 2016). East Africa is especially vulnerable to the impacts of drought (DEC, 2017). The region has long experienced widespread poverty and high levels of food insecurity (Von Grebmer et al., 2016). The high dependence of its population on rain-fed agriculture, sometimes in tandem with political instability, exacerbate the impacts of droughts (Love, 2009; Masih et al., 2014)." (Page: 1; Lines: 18-24)

*4) The absence of uncertainty estimates in Figure 1c is troubling. Most attribution studies*

*provide, at the very least, bootstrapping estimates of uncertainty. I suspect that if error bars did accompany the probability changes in the inset panel, the changes for 2001-2031 and 2035-2065 would be statistically insignificant relative to 1861-1891. Further, it is not clear as to whether the future probabilities of witnessing less than 46mm month-1 have been calculated using raw model data, or the excessively-smoothed PDF constructions.*

**[Response 2.5]** We have address all issues raised. We now calculate the bootstrapping estimates of uncertainty and add the uncertainty bounds on to the insets of Figure 1c, 1d and also on new Figure 2. The one standard deviations are estimated via bootstrapping with 80% replications from the 37 GCM precipitation data and for the 31-year periods. As the reviewer predicted, these uncertainty bounds do overlap, allowing a visual interpretation that changes may not be detectable.

We accept the reviewers' questioning of appropriateness of smoothing in our original panel 1c, and this revealed a factual error. In the revised manuscript, we now simply show bias-corrected GCM projections as a probability density function. These revised a plot (Figures 1c and 1d) – please see below.

[Figure]

**Figure 1 in the revised manuscript**. (**a**) Black rectangle is location of study region (14.5°N~1.5°S, 36°E~51°E). Plotted is mean rainfall for 2016 and months August to October inclusive (ASO), presented relative changes (as %) to long-term average ASO values (1979-

2015). Values based on ERA-interim reanalysis product. (**b**) ERA-based monthly total rainfall (mm month$^{-1}$) over study region (panel a; land within black rectangle) for years 1979 to 2016. Year 2016 is red, other years are individual grey lines, and multi-year average (not including 2016) is blue line. Blue shading is ± one standard deviation of monthly rainfall across years 1979-2015. The drought event (shaded in yellow) is defined as the three consecutive months of ASO, and when rainfall in year 2016 is below blue shading. (**c**) CMIP5-based PDFs of mean ASO rainfall for periods 1861-1891 (blue), 2001-2031 (black), 2035-2065 (orange) and 2070-2100 (red). Each curve corresponds to the mean-corrected combined outputs from 37 CMIP5 models forced by historical emissions and RCP8.5 future scenario. Individual GCM bias correction is based on the ERA-interim reanalysis product. Yellow shading is mean ASO rainfall less than 46 mm month-1, which is the ERA-interim 2016-based threshold (mean of ASO, red curve in panel b). Inset shows probabilities of mean rainfall of ASO falling below the threshold for the same modelled periods (colours match those of curves). The error bars are the standard deviations (estimated via bootstrapping 80% replications from the 37 GCM precipitation data for the 31-year periods). (**d**) same as (**c**), but based on the mean- and variance-corrected GCM rainfall estimates.

*5) The use of 1861-1891 is a misleading representation of 'pre-industrial' as it includes the influences of the Krakatoa volcanic eruption, and associated cooling effects. I strongly recommend changing the baseline period, perhaps to 1861-1880 in accordance with the IPCC's definition.*

[**Response 2.6**] We have checked this. We are keen to keep each period containing 31 years, so all statistics are comparable between timeframes analyzed. We check probability for two 31 year segments of (i) 1861-1891 and (ii) split period 1861-1881 and 1890-1899. We find for ASO East Africa rainfall, there is no difference in probabilities, and so we request keeping the pre-industrial representative years as in the original paper version.

*Reviewer #2* **Technical Comments:**
*P1, L5: East Africa, not East African.*
[**Response 2.7**] Corrected.

*P1, L10: It's not really an analysis 'merging' ERA-Interim data with CMIP5 data. Instead you are using ERA-Interim as a basis for bias-correcting the model data.*
[**Response 2.8**] We thank the reviewer pointing it out. We now have clarified this and revised

this sentence in the manuscript: "After bias correction to match contemporary rainfall mean, GCMs project small decreases in probability of drought of same severity for East Africa by the end of 21st century. However, further adjusting the variance of GCMs to match ERA-interim data, probability of drought increases slightly." (Page: 1, Lines: 9-11)

*P1, L14: '… shows that during August to October'?*
**[Response 2.9]** Corrected.

*P1, L20: RCP8.5 is not designed as a 'business-as-usual' scenario. It is in fact an acceleration of the present-day rates of increase in radiative forcing. The closest to 'business-as-usual' would be RCP6.0. I suggest the authors modify this particular phrase.*
**[Response 2.10]** To avoid confusion, the phrase of "business-as-usual scenario" is modified to "high emission future scenario".

*P1, L20-L23: This is poorly phrased. So you are bias-correcting based on climatological mean precipitation for ERA-Interim?*
**[Response 2.11]** Yes, we bias correction on climatological mean precipitation. In light of your comment 2.3 above, we also bias-correct on variance. We now re-write the method in the revised manuscript, which we hope removes previous ambiguities. We write: "A bias correction with two post-processing steps is applied to the GCM precipitation estimates. We first calculate modelled and ERA-based mean ASO rainfall estimates over the east Africa during the period 1979-2015. The GCM precipitation estimates, both past and future, are corrected by a GCM-specific mean correction factor, which is a ratio of the climatological mean of each GCM to that of the ERA-interim reanalysis product. Second, we then adjust the climatological standard deviation (STD) of GCM precipitation estimates by multiplying the ratio of the climatological STD of each GCM to that of the ERA-interim data. The adjustment of spread of rainfall distribution is an important additional procedure to further constrain GCM estimates (Sippel et al., 2016; Jeon et al., 2016; Angelil et al., 2017). Together this ensures all GCMs have the ERA-based mean and STD for period 1979-2015." (Page: 1, Lines: 28-31 and Page 2: 1-4)

*P1, L27-28: I suspect a change from 5.6 to 5.8% is not statistically significant in any way, and repeating these calculations with even just two or three models removed could lead to a completely different answer. What is the range in answers of this probabilistic increase for*

*each individual model?*

**[Response 2.12]** Following your helpful suggestion, we have answered this in two ways.

First, in response to your comment 2.5 above, we now undertake bootstrapping to provide estimates of uncertainty (Figure 1c, d; new Figure 2). These uncertainty bounds place in a much better context GCMs estimates of changes in the probability of drought occurrence in east Africa.

Second, we also calculate the changes in probabilities of drought estimated from each of individual GCMs between period 2001-2031 and period 2070-2100. This creates a new Figure 3 (please see below). For the mean-corrected GCM estimates, the changes in probabilities of drought range from -19.4% (-6 times per 31 years) to +12.9% (+4 times per 31 years). For the mean- and variance-corrected GCM estimates, drought probabilities changes range from -3.2% (-1 times per 31 years) to 12.9% (+4 times per 31 years). These values are made clear from Figure 3.

We appreciate being asked to present understanding at the individual GCM scale. This demonstrates that the choice of GCMs used is a major source of uncertainty of future drought risk analysis. Figure 3 also highlights the relatively small number of years in simulations for understanding changes in extreme events. Ideally, there would be a large ensemble of simulations by each model to refine the probability of extreme events, enabling a more complete sampling of probability distributions. Based on this, we now write in the concluding paragraph: "We find the sources of uncertainty in drought prediction include: 1) the choice of bias correction methodology; 2) the choice of observational product used to correct bias in GCMs; and 3) the choice of GCMs used." (Page: 3; Lines: 14-16) and "For rainfall of east Africa predictions in particular, this will link to accurate forward projections of oceanic variability. Strong teleconnections are known to exist between El Niño Southern Oscillation (ENSO) and East African rainfall (Segele et al., 2009; Gissila et al., 2004), and with longer-term fluctuations in Pacific SSTs increasing/decreasing rainfall (Funk et al., 2014; Liebmann et al., 2014). Larger ensembles of simulations by each model is also important, and especially when analysing the probability of extreme events. This enables a more complete sampling of probability distributions, describing more fully the internal variability of the climate system imposed over general climate change." (Page: 3; Lines: 21-26)

[Figure]

**New Figure 3 in the revised manuscript**. Changes in drought frequency, multi-year mean and standard deviations (STD) of 31 consecutive year rainfall amounts. Difference between present period 2001-2031 and period 2070-2100, as estimated by 37 GCMs. GCM estimates are corrected by the ERA-interim rainfall product. Changes to frequencies of drought occurrence are estimated from the mean bias-corrected GCM estimates (1st row), both mean- and variance bias-corrected GCM estimates (2nd row). The colored grids in the 3rd row with black borders indicate statistically significant differences in the 31-year rainfall mean between these two periods (*t*-test, with P < 0.05). The percentage changes are calculated as [($x_{2070-2100}$/$x_{2001-2031}$)-1]×100%.

*P1, L29: 'stretched left-tails' is a poor description, and I suggest changing this.*

**[Response 2.13]** We now have removed the sentences about the changes in the distribution tails in the reviewed manuscript. This is due to Review Comment 2.5.

*P2, L2-7: This is a very poor concluding paragraph. You do not summarize the key results of the study, but instead offer reasons why significance testing is needed (reliability of different models), before highlighting all of the reasons why monthly-mean precipitation deficits may in fact be a poor proxy for drought impacts over East Africa.*

**[Response 2.14]** We have completely re-written the concluding paragraph, and mainly in light of comments from both Reviewer 1 and Reviewer 2. The main difference in paper versions, is that the requested changes have placed all results in a more complete uncertainty analysis framework. We summarize these main results, noting future predictions of drought likelihood depend on the GCMs used, any bias correction algorithm, and the choice of observation product used to correct bias. Although this is a slightly negative find (i.e. answer is inclusive), we believe this is still important to have placed in the literature.

Based on Review 1 comment 1.3 above, we mention the importance of accurate representation of oceanic drivers, via teleconnections, to East Africa rainfall. In terms of drought proxy, we use the literature better in our discussion, to re-iterate that there are other factors (e.g. social drivers) of importance in terms of ability to deal with low rainfall totals.

[revised manuscript text omitted]

---

## Referee Report (RR1)

**Review for NHESS submission, "Brief Communication: Drought Likelihood for East Africa" by Yang and Huntingford**

**General comments**

I appreciate the efforts made by the authors to address my concerns, and I believe the paper has been improved significantly. However, there are still multiple issues arising from their analysis which I believe need to be taken into account, before this Brief Communication could be published.

Therefore, my recommendation is that *this article requires major revisions*. In accordance with the NHESS review criteria, my assessment is as follows:

**Scientific Significance: 3 (fair)**

Scientific Quality: 3 (fair)

Presentation Quality: 3 (fair)

\_\_\_\_\_

**Specific comments**

**1)** I'm confused by the presentation of results in Figure 2. The authors adjust each model according to a different observational data set, and then proceed to look at the likelihood of the 2016 drought (or worse) in each new ensemble. But is the drought event threshold for each of these new ensembles still 46mm? Or is it the absolute rainfall total associated with ASO 2016 for each individual dataset? It is unclear based on the current text. Also, if it is the latter approach, this is equally problematic, since the sigma-anomaly associated with the ASO2016 event might differ dramatically between the different observational products, which would thus render Figure 2 as no longer an apples-for-apples approach.

My recommendation would be to identify the percentile anomaly associated with ASO2016 for each observational product, then take the mean of these answers, and use this average percentile anomaly as the event threshold employed for all panels in both Figures 1 and 2.

**2)** As it stands, the authors first bias-correct the mean, and sometimes then also variance, for each individual model, before combining all models into a single ensemble and calculating the relevant statistics. How do the answers differ if you instead combine all raw model results into a single ensemble first, and then proceed to correct the multi-model ensemble by a singular correction factor for mean (and then variance)?

**3)** A paper recently published has provided a multi-method assessment of changes to the likelihood of occurrence of the Ethiopian drought of 2015 (doi: 10.1175/JCLI-D-17-0274.1). The authors will therefore need to provide explicit justification of the *value added* by publishing their analysis, relative to this already-published paper. Especially given the significant level of overlap in both regions considered and topics discussed (specifically, changes in drought likelihood in the context of ongoing climate change).

Second, the CMIP5-relevant part of the 2015-relevant analysis employs a specific method of event attribution, whereby they exclude models from subsequent analysis of 'changes in drought likelihood' if the rainfall climatology of a given model did not match the climatology from observations (using a KS test). Given this CMIP5-based approach has also been used in multiple other attribution studies, I would like the authors to comment on how their approach is better or worse, and why. For example, if you employed this alternative approach to your analysis, how would the answers change?

4) There are two additional observation-based precipitation products which would be of considerable value to add to this analysis: CRU-TS (as I previously highlighted), and CenTrends dataset (doi:10.1038/sdata.2015.50), which has been specifically developed for analysis of seasonal rainfall anomalies over the Horn of Africa region. Further, based on these updated, observation-based products, I would strongly question the continued use of ERA-Interim as the primary product of consideration in Figure 1.

**5)** The bottom row of Figure 3 presents changes to standard deviations in model rainfall, based on running 31-year periods. It's not clear to me what this standard deviation represents: if it's based on '31-year rainfall mean' as stated, then this implies only one data point. Or is it the annual average for each of the 31 years, or the ASO-averaged rainfall per year? And besides, either of these latter suggestions would yield only 31 data points, hence I'm not sure it provides any information of real value. This is particularly true, given the fact that a purported 121% increase in rainfall SD (by MIROC5) is not considered a statistically significant increase.

My recommendation is to remove the row considering future changes in model variability, and just leave the row mentioning changes in the mean.

**6)** I have significant issues with the treatment of uncertainty estimates in Figures 1 and 2. The width of the error bars is an implicit estimate of whether a 'statistically significant' difference exists. By showing error bars with 1-standard deviation only, this implies a 68% confidence level (using the assumption of a normal distribution). Most studies tend to consider uncertainty estimates based on confidence levels of 90% or higher. I strongly recommend presenting all uncertainty ranges on the figures using two standard deviations – this will be more representative as to whether a statistically significant difference really exists or not.

---

## Author Response (AR2)

December 2017

Dear Editor of NHESS

Thank you for your on-going help with this manuscript, and we are grateful that the two reviewers were prepared to look again at our Brief Communication:

**Drought Likelihood for East Africa.**

The main technical change is the request to include in the analysis two additional measurement datasets. These are the CRU-TS and a member of the CenTrends dataset family (CHIRPS). This now leads to two additional panels in Figure 2, where we normalize climate models against different rainfall datasets. From these, we calculate future risk of drought.

We have also adopted the request to present all of Figure 1, which captures the magnitude of the Autumn 2016 East Africa drought, in terms of the CenTrends data (CHIRPS).

All other technical requests we implemented in full, and outlined in our responses below. The one exception is the query as to whether our normalization of GCMs to each alternative measurement dataset is a like-for-like comparison. We believe that it is, but recognize that the wording of the paper in the original version needed to be improved to remove any ambiguity.

Our responses are described below, and including presentation of adjusted text. We realise that both reviewers have requested significant changes through the progression of this paper. However we hope that the manuscript is now acceptable as a short summary document capturing current projections of autumn rainfall by GCMs for East Africa. This is where the severe drought of 2016 occurred, enabling assessment of future likelihood of reoccurrence.

Thank you for all the help so far. With kind regards,

Hi Yag

Hui Yang (and on behalf of Chris Huntingford). Email: yang\_hui@pku.edu.cn

**Response to the reviewers**

**To Reviewer #1:**

**Reviewer #1** General Comments:**

The analysis and description of the method in this study has been significantly improved during the revision. The additional figures and tables help the reader understand the method and the results. The conclusion has been significantly improved and it now puts the study and its results in context. This study can be published after several small technical corrections, listed below. [Response] We are pleased that the reviewer likes our new paper version. Thank for all your comments and suggestions that have helped us generate a better manuscript.

**LANGUAGE: The English could still be improved in several places.**

Abstract: The abstract still has to be improved in terms of clarity and language, especially the second part.

**[Response]** We have polished the language in large parts of manuscript. We hope the new paper version has a level of clarity as to be useful to a relatively broad audience. The uploaded version to the NHESS website is with "track-changes" so our additional minor edits can also be seen.

**Reviewer #1** Detailed Comments:**

Page 1, line 31: an equation explaining how this is computed would be helpful, e.g. is daily data used?

**[Response 1.1]** We have followed the reviewer's advice and now put the bias-corrected equations in the main text. We use indexing and superscript notation to now make it unambiguous as to exactly the correction we apply. We write: "The GCM precipitation mean ASO estimates, both past and future, are corrected for each model year by a GCM-specific mean correction factor. This factor is a ratio of the climatological mean of each GCM to that of the CHIRPS product as:

$$x_{corr,i,j}^{\mu} = x_{\text{mod}\,el,i,j} \times \frac{\mu_{obs}}{\mu_{\text{mod}\,el,j}} \quad (1)$$

Here  $x_{\text{mod}el,i,j}$  and  $x_{corr.i.j}^{\mu}$  are, respectively, model simulated and mean bias-corrected ASO precipitation data of the *i*th year (*i*=1,2,...,31) for the *j*th GCM (*j*=1,2,...,37).  $\mu_{obs}$  and  $\mu_{\text{mod}el,j}$  are the observed and GCM-specific time-mean (i.e. average across indices *i*) of ASO rainfall estimates during the period 1981-2015. Second, we then adjust the mean-corrected data from

Eqn. (1), such that they further are corrected to have identical standard deviation (STD) to the CHIRPS product whilst maintaining the bias correction for the mean. This gives bias-corrected estimates  $x_{corr,i,j}^{\mu,\sigma}$  as:

$$x_{corr,i,j}^{\mu,\sigma} = (x_{corr,i,j}^{\mu} - \overline{x_{corr,j}^{\mu}}) \times (\frac{\sigma_{obs}}{\sigma_{corr,j}^{\mu}}) + \overline{x_{corr,j}^{\mu}} \quad . \tag{2}$$

Here  $\overline{x_{corr,j}^{\mu}}$  (=  $\mu_{obs}$ ) is the 31-year average of mean bias-corrected data from Eqn. (1).  $\sigma_{obs}$  and  $\sigma_{corr,j}^{\mu}$  are standard deviation of the ASO rainfall estimates during the period 1979-2015 from observations and from the mean bias-corrected precipitation data created by Eqn. (1)." (Page: 2, Lines: 1-14)

Page 1, line 29 / page 2, lines 19 and 25: change "the east Africa" to "East Africa" or "eastern Africa"

**[Response 1.2]** Done at the noted points in the manuscript. We thank the referee for pointing this out.

Page 3, line 23: the papers listed below might be helpful references

*Gleixner, S., Keenlyside, N., Viste, E., & Korecha, D. (2016). The El Niño effect on Ethiopian summer rainfall. Climate Dynamics, 49(5-6), 1865–1883.* http://doi.org/10.1007/s00382-016- 3421-z

Gleixner, S., Keenlyside, N. S., Dimissie, T., Counillon, F., Wang, Y., & Viste, E. (2017). Seasonal predictability of Kiremt rainfall in CGCMs. Environmental Research Letters. http://doi.org/10.1088/1748-9326/aa8cfa

[**Response 1.4**] Thank for this. We have cited this paper quoted by the reviewer in the line \*\*. We now write: "Strong teleconnections are known to exist between El Niño Southern Oscillation (ENSO) and East African rainfall (Segele et al., 2009; Gissila et al., 2004; Gleixner et al., 2016), and with longer-term fluctuations in Pacific SSTs either increasing or decreasing rainfall (Funk et al., 2014; Liebmann et al., 2014; Gleixner et al., 2017)." (Page: 5; Lines: 1-3)

**Page 7, Figure 2, caption: please explain better**

**[Response 1.5]** We re-write the caption of Figure 2, and the revised text now reads: "**Figure 2**: CMIP5 GCM-based histograms of probabilities of mean ASO rainfall falling below year 2016-based threshold values. This is for different time periods and for different observation-based precipitation products of CHIRPS, CRU-TS, ERA-interim, GPCP, PREC/L, CPC and TRMM.

Shown for years 1861-1891 (blue), 2001-2031 (black), 2035-2065 (orange) and 2070-2100 (red), and using GCMs simulations corresponding to historical and RCP8.5 estimates. Individual GCM projections are bias-corrected by the (panel-specific) precipitation product. This data is combined to give single overall probabilities across the 37 GCMs sampled. The histogram bars without horizontal hatching (left) are for the mean-corrected GCM precipitation estimates. The bars with hatching (right) are for the mean- and variance-corrected GCM estimates. The error bars are two standard deviations (estimated via bootstrapping 80% replications from the 37 GCM precipitation data for the 31-year periods). Data in the CHIRPS panels repeats that of the insets of Figs 1c, d." (Page: 9)

**To Reviewer #2:**

**Reviewer #2** General Comments:**

I appreciate the efforts made by the authors to address my concerns, and I believe the paper has been improved significantly. However, there are still multiple issues arising from their analysis which I believe need to be taken into account, before this Brief Communication could be published.

Therefore, my recommendation is that this article requires major revisions.

**[Response]** We thank the reviewer for re-reviewing the manuscript and providing us with an additional set of helpful comments. This insightful advice has helped us to improve the manuscript further. Our responses are below, and revised text in the paper repeated in red font.

**Reviewer #2* Specific Comments:**

1) I'm confused by the presentation of results in Figure 2. The authors adjust each model according to a different observational data set, and then proceed to look at the likelihood of the 2016 drought (or worse) in each new ensemble. But is the drought event threshold for each of these new ensembles still 46mm? Or is it the absolute rainfall total associated with ASO 2016 for each individual dataset? It is unclear based on the current text. Also, if it is the latter approach, this is equally problematic, since the sigma-anomaly associated with the ASO2016 event might differ dramatically between the different observational products, which would thus render Figure 2 as no longer an apples-for-apples approach.

My recommendation would be to identify the percentile anomaly associated with ASO2016 for each observational product, then take the mean of these answers, and use this average percentile anomaly as the event threshold employed for all panels in both Figures 1 and 2.

**[Response 2.1]** The threshold is different for each observational record. It is based on that dataset and its estimate of ASO rainfall in 2016. These values are given in Table S2. The number 46mm is for ERA-interim, however your request (below) to use CHIRPS data, the threshold in that instance is 40mm. (Hence we use the second option the reviewer notes). We state this more clearly in the paper, now writing "The probability of drought occurrence is based on estimates of ASO rainfall in 2016 and for each individual dataset (values in Table S2)."

Each measurement dataset does have different distance between the year 2016 ASO rainfall value and the ASO climatological mean, different climatological mean and different climatological standard deviation (see Table below, which also appears as Table S2). That is,

based on different product, the severity of year 2016 ASO drought are different. We then correct both mean and standard deviation of combined GCMs estimates, and for climatological period (1981-2015), to be identical to each dataset. With both these statistics ( $\mu$  and  $\sigma$ ) identical, we believe that for the historical period, the model estimates corrected by different precipitation data are like-for-like. From this, we assess future model projections in the context of how rainfall below the year 2016 ASO threshold is expected to change. Hence such changes are GCM and dataset specific. We have considered at length this, and think this is an appropriate like-for-like ("apples-for-apples"), but recognize the original paper wording could have been made clearer in this regard.

We now write "Large uncertainty in the observation-based precipitation products has been well reported (Angélil et al., 2016), we additionally use six other precipitation estimates (CRU-TS, ERA-interim, GPCP, PREC/L, CPC and TRMM) to bias-correct GCM estimates. The probability of drought occurrence is based on estimates of ASO rainfall in 2016 and for each individual dataset (values in Table S2). There are substantial differences between these values. We use each of these extra datasets to repeat the bias-correction of every GCM by same Eqn. (1) and Eqn. (2), but now with new data-specific  $\mu$  and  $\sigma$  values. These  $\mu$  and  $\sigma$  quantities are also given in Table S2." (Page: 3, Lines: 6-11)

**Table S2**. The mean August-to-October (ASO) rainfall (mm month-1) of year 2016, multi-year mean (not including 2016) and multi-year standard deviation (STD, not including 2016) over east Africa for years 1981 to 2015. The seven global precipitation data sets used are listed. Six products of CHIRPS, CRU-TS, ERA-interim, GPCP, PREC/L, CPC and TRMM are available from 1981 to 2016. These six precipitation data sets are either interpolated gauge observations only (i.e. CHIRPS, CRU-TS, PREC/L and CPC), gauge observations combined with satellite measurements (i.e. GPCP), or reanalysis data (i.e. ERA-interim). The TRMM satellite observations are available from 2001 to 2016.

| ASO rainfall                                   | CLUDDS | CRU-  | ERA-    | CDCD  |        | CDC   |        |
|------------------------------------------------|--------|-------|---------|-------|--------|-------|--------|
| (mm month -1 )                      | СПІКРЗ | TS    | interim | GPCP  | PREC/L | CPC   | IKIM   |
| 2016                                           | 39.97  | 45.93 | 46.10   | 46.56 | 57.16  | 35.78 | 32.05  |
| Climatological
mean, μ
(1981-2015)       | 53.05  | 55.49 | 70.81   | 62.94 | 62.01  | 44.59 | 60.69* |
| Climatological
STD, $\sigma$
(1981-2015) | 7.09   | 10.20 | 11.55   | 10.68 | 11.72  | 13.33 | 11.83* |

\* TRMM satellite precipitation data is only available from 2001 to 2016. The climatological ASO rainfall averages of the period 2001-2015 is computed.

2) As it stands, the authors first bias-correct the mean, and sometimes then also variance, for each individual model, before combining all models into a single ensemble and calculating the

relevant statistics. How do the answers differ if you instead combine all raw model results into a single ensemble first, and then proceed to correct the multi-model ensemble by a singular correction factor for mean (and then variance)?

**[Response 2.2]** Thank you for this. We perform this analysis as the reviewer suggests i.e. instead of normalizing the models individually, we instead normalize just once the full ensemble of GCM estimates. That is, we use  $\mu$  and  $\sigma$  from a single array that contains all years and from all GCMs. This generates a diagram identical in format to Figure 2, which we put in SI, and as repeated below.